# Multiscale stress dynamics in sheared liquid foams revealed by tomo-rheoscopy

Florian Schott [1] ✉, Benjamin Dollet[2,8], Stéphane Santucci [3,8] ✉,
Christian Matthias Schlepütz [4], Cyrille Claudet[5], Stefan Gstöhl [4],
Christophe Raufaste [5,6,8] & Rajmund Mokso[1,7,8]

Rheology aims at quantifying the response of materials to mechanical forcing. However, standard rheometers provide only global macroscopic quantities, such as viscoelastic moduli. They fail to capture the heterogeneous flow of soft amorphous materials at the mesoscopic scale, arising from the rearrangements of the microstructural elements, that must be accounted for to build predictive models. To address this experimental challenge, we have combined shear rheometry and time-resolved X-ray micro-tomography on 3D liquid foams used as model soft jammed materials, yielding a unique access to the stresses and contact network topology at the bubble scale. We reveal a universal scaling behavior of the local stress build-up and relaxation associated with topological modifications. Moreover, these plastic events redistribute stress non-locally, as if the foam were an elastic medium subjected to a quadrupolar deformation. Our findings clarify how the macroscopic elasto-plastic behavior of amorphous materials emerges from the spatiotemporal stress variations induced by microstructural rearrangements.

Gels, pastes, emulsions, foams and colloidal suspensions are typical examples of soft amorphous materials that we commonly use in our daily lives for their unique mechanical properties, exhibiting both solid-like and liquid-like behavior. This dual nature is indeed at the root of their use in numerous applications and industrial processes, spanning sectors such as food, cosmetics, pharmaceuticals, building materials, resource extraction, and environmental remediation[1–3]. Their elementary constituents —such as bubbles, droplets, or grains— are jammed, meaning they are densely packed and constrained by their neighbors. In this jammed state, the material behaves like a rigid elastic solid, capable of sustaining and transmitting stresses through its disordered structure. Above a critical stress, these disordered materials undergo a yielding transition from a solid to a liquid state due to irreversible topological rearrangements at the microscopic scale of their jammed constituents. Despite their widespread use with

numerous commercial products and extensive research efforts, a fundamental understanding and control of this unjamming transition remains a formidable challenge today[4,5]. This difficulty arises from the inherent complexity in directly observing and characterizing the out-of-equilibrium evolution of the network topology of the material constituents. Indeed, standard rheological tools provide only a global macroscopic mechanical response of the probed material. Thus, they lack the spatio-temporal resolution necessary to capture the heterogeneous mechanical flows of those soft disordered materials, characterized by correlated structural rearrangements that lead to slips, shear bands, or fractures[6–8].

In recent years, significant progress has been reported[9] thanks to the use of different optical techniques such as confocal microscopy[10–12], light scattering[13], or differential dynamic microscopy[14,15] coupled to rheometry, allowing to observe the microscopic dynamics of soft

[1]Division of Solid Mechanics, LTH, Lund University, Lund, Sweden. [2]Université Grenoble Alpes, CNRS, LIPhy, Grenoble, France. [3]CNRS, ENS de Lyon, LPENSL, UMR5672, Lyon cedex 07, France. [4]Swiss Light Source, Paul Scherrer Institut, Villigen, Switzerland. [5]Université Côte d'Azur, CNRS, INPHYNI, Nice, France. [6]Institut Universitaire de France (IUF), Paris, France. [7]Department of Physics, Technical University of Denmark, Kgs. Lyngby, Denmark. [8]These authors jointly supervised this work: Benjamin Dollet, Stéphane Santucci, Christophe Raufaste, Rajmund Mokso. ✉e-mail: florian.schott@solid.lth.se; stephane.santucci@ens-lyon.fr

jammed materials, and particle rearrangements occurring during shear tests[16–18]. However, these methods present strong limitations. In particular, confocal microscopy requires the use of (fluorescent) colloidal particles - thus most of current experiments concern only colloidal glasses, while Fourier-based techniques do not provide any direct information on individual particles. More importantly, none of these techniques can offer a measure of the stress at the local scale of the jammed constituents of the materials.

Interestingly, liquid foams, archetype of soft jammed materials[19], possess remarkable properties that make them ideal candidates for characterizing stress down to the scale of their inner structure, specifically the bubbles[20]. First of all, the difference in density between the gaseous and liquid phases can be exploited as an intrinsic contrast agent for 3D imaging technique such as X-ray tomography[21–25]. New developments in X-ray imaging allowed recently to probe foam dynamics below the time resolution of one second, with notably a 3D image (a tomogram reconstructed from thousands of projections) acquired in less than 0.5 seconds[26]. Second, the stress at the local scale of a single bubble can be inferred from the bubble's shape through the intermediate of surface tension, which was already exploited in simulations[27,28], but never in 3D experiments. Furthermore, modifying the physico-chemistry of the foaming solution and the nature of the bubble gas, one can obtain ultra-stable foams. In these foams, the ageing processes—such as liquid drainage, bubble coalescence, and coarsening—can be significantly slowed down[29], and the duration of a topological rearrangement can be controlled over a large range of time scales[30].

Therefore, we propose here an original experimental study on stable liquid foams, where we combine simultaneous time-resolved fast X-ray micro-tomography with continuous rheological shear testing in a plate-plate geometry. Our tomo-rheoscopy allows to overcome current state-of-the-art limitations (low spatial and temporal resolution, reduced field of view, turbidity/opacity of the material) to directly observe and monitor the local deformation, stress, and contact topology of each bubble within the sheared foams. First, by analyzing the shape of each segmented bubble (up to $10^5$ in a single tomogram) within the sheared foam, we obtain the deviatoric stress tensor at the scale of each individual bubble. Computing the corresponding stress field over larger volumes up to the entire foam sample leads to a measurement of the shear stress applied on the rheometer plates, which is in quantitative agreement with the independently measured torque provided by the rheometer, thus validating our local stress measurements, derived from image analysis of the reconstructed bubble film shapes. Then, by tracking the contact network topology of each bubble, we are able to unveil the detailed spatio-temporal mechanical response of the sheared foam at the bubble scale. We focus on elementary topological rearrangements, known as T1 events[31], which consist of the loss of contact between two bubbles and the simultaneous creation of a new contact between two neighbouring bubbles. Such neighbour-swapping events involving four bubbles represent the elementary plastic process for foams. Our tomo-rheoscopy of liquid foams allows us to reveal a universal scaling behavior in local stress build-up and relaxation associated with such elementary plastic events, from which the macroscopic visco-elasto-plastic behaviour of the foam emerges. Indeed, for the various foams probed, with different liquid fractions, and subjected to different local shear rates, we obtain a master curve describing the local stress as a function of the local strain. On average, each bubble involved in a T1 event exhibits a stress increase before the T1, with bubbles losing and gaining contact showing increases of 20% and 10% above the yield stress, respectively. This is followed by a stress drop of 40% below the yield stress after the T1 in both cases. Furthermore, we also demonstrate that these elementary plastic events spatially redistribute stress in a non-local manner, as if the foam were an elastic medium subjected to a quadrupolar deformation, in accordance with Eshelby's prediction[32].

Our original experimental setup and analysis demonstrate that the fast tomo-rheoscopy of a model soft-jammed material can offer a complete and comprehensive understanding of the yielding of the material's constitutive structural elements within a shear flow. This breakthrough opens up an entirely new avenue for research, calling for new experimental campaigns to study various types of liquid foam samples[33] and, more broadly, other soft jammed materials submitted to different flow conditions, such as oscillatory or creep tests to uncover the fundamental causes and origin of their macroscopic rheological properties.

## Results

Eight liquid foam samples were prepared as detailed in Methods. They are almost monodisperse, with an average radius varying across samples between 50 and 80 $\mu$m, and a liquid fraction between 5% and 21% constant in space and time within a few percents (see Supplementary Information). Their stability was optimised such that no significant ageing could be measured during the experimental runs.

In order to perform simultaneously continuous shear testing and time-resolved X-ray micro-tomography imaging of such liquid foam samples, we have designed and developed an innovative set-up based on a dual-motor rheometer prototype, described in Methods. We use here a plate-plate geometry of radius $R = 2.75$ mm and gap $h = 1.5$ mm, with rough plates ensuring no-slip boundary conditions (Fig. 1a). The plates are in relative rotation at a low angular speed $\Omega = 2\pi \times 10^{-3}$ rad/s. The plate-plate geometry allows for the application of different local shear rates. The local shear rate component in the $\theta z$ plane $\dot{\gamma}(r) = \Omega r/h$ increases linearly with the radial distance $r$ from the rotation axis, considering a cylindrical coordinate system $(r, \theta, z)$ centered on the rotation axis, with $z = 0$ corresponding to the lower plate. We verified that the flow field follows closely the applied shear rate (see Methods): no shear banding is present in our experiments.

Three-dimensional (3D) images (called tomograms) are acquired every 3 s using the fast X-ray tomography facility of the Swiss Light Source. Each experiments counts 160 tomograms, imaging the liquid foam sample over a volume of around 30 mm³ with $2016 \times 2016 \times 800$ voxels and a voxel size equal to 2.75 $\mu$m. A typical tomogram is shown in Fig. 1b. Movies of typical experiments in various cross-sections are also provided in Supplementary Information. Bubbles are segmented as detailed in Methods, as shown in Fig. 1c. The thickness of the bubble films in our liquid foams is significantly smaller than the voxel size in our tomographic images. As a result, although our spatial resolution is sufficient to image the network of interconnected liquid channels—on the scale of several tens of micrometers—one cannot directly observe or detect the thin bubble films. However, we have previously developed and validated a phase and bubble segmentation and reconstruction procedure based on watershed analysis[22,24,34]. On the other hand, the relatively limited temporal resolution was chosen as a compromise between computational memory constraints and the targeted duration of the experiments (approximately 10 minutes), while limiting radiation exposure to avoid altering the liquid foam's physical chemistry.

Our experiments are run as follows. The liquid foam is placed within the rheometer gap, using a syringe. In order to reduce subsequent residual bubble deformation, before each test, we first apply a pre-shear with a high relative rotation speed of 50 $\Omega$ for 20 s in one direction followed by a shear in the opposite direction at the same rate for an equal duration. The sample is let to relax for 80 s, and then sheared at $\Omega$ while simultaneously imaged. Our shear experiments last 480 s. This procedure yields a unique and unprecedented set of experimental data, with an extensive statistics, giving access to the position, shape, contact topology of each bubbles within sheared liquid foams, while measuring the temporal evolution of the global shear stress by the rheometer[26]. As explained in the next section, based

on the analysis of the shape of the bubble surface, one can moreover obtain a measurement of the stress at the local bubble scale.

## Multi-scale stress measurements – *from the bubble to the foam scale*

At low enough shear rates, viscous stresses are negligible, and the only contribution to the stress is associated with the interfaces through the surface tension $\Gamma$. The non-isotropic part of the stress is given by $\sigma_{ij} = \frac{\Gamma}{V} \int (\frac{1}{3}\delta_{ij} - n_i n_j) \, dS$ [28,35,36], where integration is carried on all interface elements $dS$, with unit normal vector **n** of components $n_i$, contained in a volume $V$; $\delta_{ij}$ is the Kronecker unit tensor, $\delta_{ij} = 1$ if $i = j$ and 0 otherwise. Therefore, meshing the interface of each labelled bubble of our sheared foams (marching cube), with triangular surface elements, as illustrated in Fig. 2a, we could compute this stress tensor at the bubble scale of volume $V$, by performing an integration over all those triangular surface elements $dS$, of normal unit vector **n**.

Then, the macroscopic stress in larger volumes containing numerous bubbles is obtained by averaging the stress of individual bubbles whose centroids lie in such volumes. It is expressed in cylindrical coordinates inside tori to account for the axisymmetric geometry. One torus is defined by the volume containing all the bubbles between $r$ and $r + \Delta r$. In practice, the radial axis is divided into 10 sections that typically contain 84–345 and 1970–6987 bubbles for the smallest and largest radii, for the largest and smallest bubble size, respectively.

We first consider this upscaling procedure for a shear experiment of a dry foam, with a liquid fraction of $\phi_\ell = 8.4\%$, composed of bubbles of average radius of 52 $\mu$m (series 2 in Table 1). As expected with the geometry of our set-up, the relevant and main stress component is $\sigma_{\theta z}$ during our shear experiments, even though one can also notice the emergence of non-negligible normal stress components $\sigma_{\theta\theta}$ and $\sigma_{zz}$ (see the Supplementary Information, for the various components of the stress tensor). We plot the time evolution of the stress component $\sigma_{\theta z}$ at various distances $r$ from the rotation axis in the inset of Fig. 2b. This stress component first increases before reaching a plateau. The initial increase is sharper for larger values of $r$, since the local shear rate component in the $\theta z$ plane $\dot{\gamma}(r) = \Omega r/h$ increases linearly with $r$. Accordingly, plotting $\sigma_{\theta z}$ as a function of the local amplitude of the applied deformation $\gamma = \dot{\gamma}(r)t$ rescales extremely well all our data (Fig. 2b). The trend of this master curve is classical for a yield stress fluid. Specifically, all data follow a unique elastic loading (with a slope giving the elastic modulus $G$) until a steady-state strain $\gamma_{ss}$, above which the stress saturates at a steady-state shear stress value $\sigma_{ss}$. The precision of our measurements reveals a slight effect of the local shear rate $\dot{\gamma}$ on this stress amplitude $\sigma_{ss}$: the larger the shear rate, the larger the steady-state shear stress; nevertheless, these variations are minimal. At most, there is a 20% increase of the steady-state shear stress when the shear rate is tripled, indicating that the liquid foam experiences a quasi-static flow. One can also notice that this shear stress value $\sigma_{ss}$ exhibits a slight drift, which can be attributed to a small yet measurable

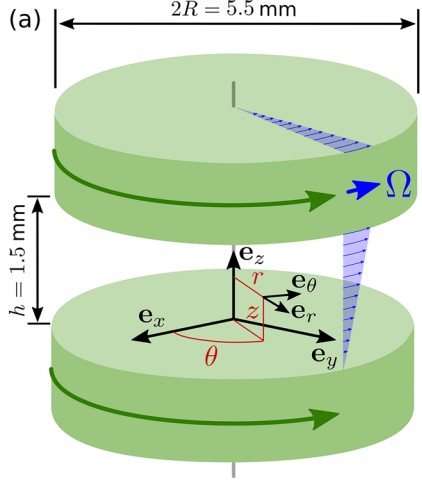

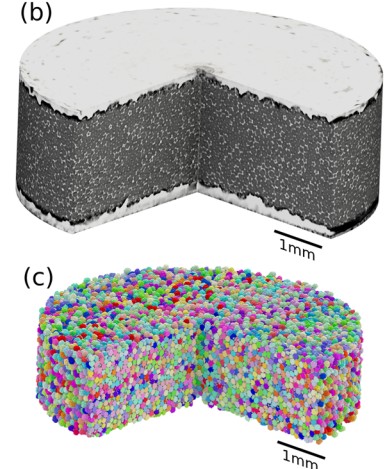

**Fig. 1 | Tomo-rheoscopy and image analysis. a** Sketch of the co-rotation rheometer geometry allowing simultaneous continuous shear testing and microtomography imaging. **b** Typical 3D raw image with a quarter vertical cut showing the bubbles within the gap. **c** Corresponding bubble segmented image. The bubbles are distinguished by their color.

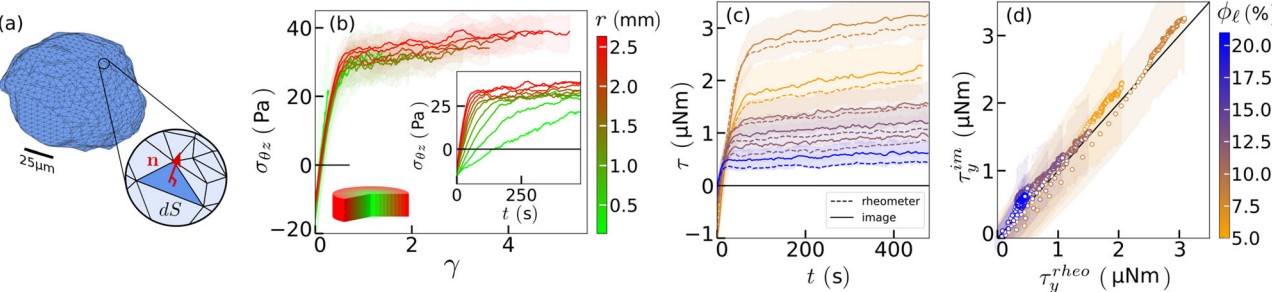

**Fig. 2 | Multi-scale stress measurements. a** Example of a meshed bubble interface obtained by marching cube, over which the Batchelor stress tensor is integrated with each individual triangle surface $dS$ and normal unit vector **n**. **b** The shear stress $\sigma_{\theta z}$ at various radius rescales when shown as a function of the local deformation $\gamma$. $\sigma_{\theta z}$ as a function of time is shown in the sub-figure. **c** Torque measured by the rheometer and obtained from image analysis as a function of time for various series. **d** Torque derived from image analysis plotted against the corresponding rheometer values for the same data as in **c**.

**Table 1 | Liquid fraction and average bubble radius for the eight series**

| Series | $\phi_\ell$ (%) | $R_{32}$ (μm) |
|---|---|---|
| 1 | 5.7 | 86.9 ± 4.3 |
| 2 | 8.4 | 57.8 ± 1.0 |
| 3 | 11.0 | 67.6 ± 0.9 |
| 4 | 12.2 | 84.6 ± 4.0 |
| 5 | 13.0 | 68.0 ± 1.4 |
| 6 | 14.6 | 85.7 ± 4.8 |
| 7 | 18.6 | 68.0 ± 0.6 |
| 8 | 21.7 | 54.0 ± 0.7 |

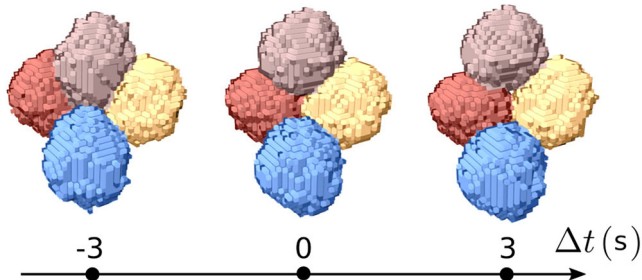

**Fig. 3 | Direct observation of a T1 event in three dimensions.** Typical example of a detected T1 event over three successive time steps.

decrease in the foam's liquid fraction over the course of the experiment (as shown in Supplementary Information). Nevertheless, such data obtained from our local stress measurements give access to the macroscopic intrinsic rheological properties of our liquid foams, such as the elastic modulus $G$, the yield stress $\sigma_Y$ and the yield strain $\gamma_Y = \sigma_Y/G$. Indeed, the yield stress $\sigma_Y$ can be obtained through an affine extrapolation of the steady-state shear stress $\sigma_{ss}$ in the limit of shear rates approaching zero, $\dot{\gamma} \to 0$ (see Methods). We show notably that our measurements are in quantitative agreement with those obtained from classical rheological measures[37,38], and notably their evolution with the liquid fraction.

Finally, we can compute the macroscopic stress at the foam scale (based on our fast imaging and detailed analysis at the local bubble interface scale) and compare such measurement to the independently measured torque $\tau^{\text{rheometer}}$ provided by the rheometer. Indeed, integrating the local stress exerted by the foam on the rotating plate yields the following prediction: $\tau^{\text{image}} = 2\pi \int_0^R r^2 \sigma_{\theta z}|_{z=h}\, dr$. We plot the time evolution of $\tau^{\text{rheometer}}$ and $\tau^{\text{image}}$ for foams of different liquid fractions in Fig. 2c and d. Remarkably, our global torque estimation based on the analysis of the shape of the bubbles' surfaces is in excellent quantitative agreement with the independent measurement given by the rheometer, for our various shear experiments performed over different liquid foams, for which we systematically varied their liquid fraction. Furthermore, we provide in Supplementary Information a complementary rheological characterization of our liquid foams, presenting flow curves, typical of yield stress fluids, that follow, Herschel-Bulkley model: $\sigma_{ss} = \sigma_Y + K\dot{\gamma}^n$, with an exponent $n$ close to 0.25. This shear thinning behaviour is in quantitative agreement with the literature[39], and the steady-state stress measurements obtained by our tomo-rheoscopy analysis at low shear rates.

Therefore, we demonstrate here our ability to measure and monitor the stress from the local bubble scale up to the global foam scale, during a quasi-static shear test. We can now take advantage of such local stress measurement to study the plastic properties of our liquid foams.

## Universal mechanics of an elementary topological rearrangement

We can track and monitor the temporal evolution of the contact network topology of each bubble in the sheared liquid foams. More specifically, we can detect topological changes between consecutive images that correspond to the loss or creation of contacts between bubbles. Here, we propose to focus solely on the so-called T1 events, the elementary topological rearrangements involving the simultaneous loss of a contact between two bubbles and the creation of a new contact between two other neighbouring bubbles. For a given radial distance, which corresponds to a given shear rate, we recorded between 5000 and 33,000 T1 events for the smallest and largest radii, respectively, considering only the events occurring on the plastic steady-state shear stress plateau. Figure 3 presents a typical example of a detected T1 event, illustrating the evolution of the four bubbles involved in this elementary topological rearrangement over three successive time steps. In the Supplementary Information, we also include a figure indicating the location and orientation of this T1 event within the sheared foam, along with a movie showing the event over a longer period of time.

We plot in Fig. 4a and b the averaged shear stress component $\sigma_{\theta z}$ of the bubbles losing a contact, as a function of time $\Delta t$, with the convention that $\Delta t = 0$ marks the occurrence of the topological change. The time evolution of this quantity $\sigma_{\theta z}$ is studied as a function of the local shear rate (corresponding to different radial distances $r$) for a given liquid fraction (Fig. 4a) or as a function of the liquid fraction for a given local shear rate (Fig. 4b, c). Interestingly, our data extracted from very different experimental conditions display the same behaviour: far from the T1 event, the averaged shear stress component $\sigma_{\theta z}$ takes a value compatible with the plastic steady-state shear stress plateau $\sigma_{ss}$, previously quantified for all bubbles at the foam scale (Fig. 2b). Then, $\sigma_{\theta z}$ increases until it reaches a maximum just before the T1 event, when it sharply decreases to a minimum value, lower than the plastic plateau, marking the post-T1 relaxation. It then increases anew before recovering the value of the plastic plateau. One can notice that the larger the shear rate or the liquid fraction, the faster the variations associated with the imposed deformation. The stress relaxation duration $\Delta t_{T1}$ associated with a T1 event (measured as the time difference when reaching the corresponding maximum and minimum shear stress values) varies from 7 to 12 s, depending on the liquid fraction of the probed foam (Fig. 4f). Such a time scale is consistent with measurements performed over a sheared cluster of four bubbles, formulated with similar rigid surfactants solutions, and shown to be controlled by the ratio between surface tension and interfacial viscous forces[30]. This characteristic time is significantly shorter than the timescale $1/\dot{\gamma}$ of the imposed deformation, making our shear experiments effectively quasi-static.

Remarkably, all data collapse onto a master curve when $(\sigma_{\theta z} - \sigma_{ss})/\sigma_{ss}$ is plotted as a function of $\Delta\gamma/\gamma_{ss}$ (Fig. 4d), with $\Delta\gamma = \dot{\gamma}\Delta t$ and $\gamma_{ss} = \sigma_{ss}/G$. This data collapse (including measurements for various shear rates and liquid fractions) is obtained thanks to the use of the macroscopic foams properties $\sigma_{ss}$ and $\gamma_{ss}$. Such a result highlights that the macroscopic visco-elasto-plastic behaviour of our 3D liquid foams reflects the local mechanical properties of the individual bubbles, and specifically the local stress variations associated with their structural rearrangements. Quantitatively, variations occur within a range of $\Delta\gamma = \pm\,\gamma_{ss}$, where the maximum stress represents a 20% increase with respect to the steady-state shear stress value $\sigma_{ss}$, followed by a subsequent peak-to-peak stress drop of 40%.

Strikingly, the same phenomenology and a similar data collapse were obtained when analysing the stress variations for the two bubbles gaining contacts during the same T1 events. Nevertheless, in this case, the stress increase is only 10%, marking a difference that could be exploited to distinguish and sort the behaviour of those yielding bubbles. The post-T1 relaxation for the bubbles gaining a

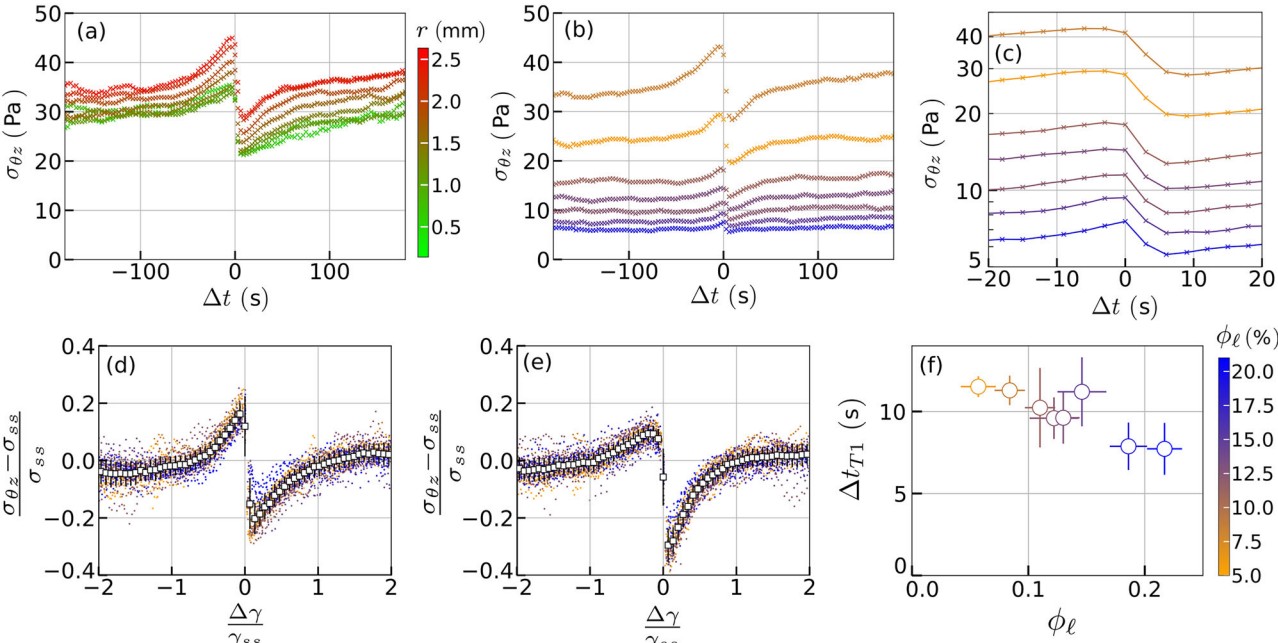

**Fig. 4 | Universal mechanical signature of a T1 event. a** Time evolution of the averaged shear component $\sigma_{\theta z}$ for bubbles losing a contact during the T1 event at different shear rates in the range 0.003–0.011 s⁻¹, corresponding to different radial distances from the rotation axis within the rheometer gap. **b** The same approach for experiments with different liquid fractions and a fixed shear rate of 0.01 s⁻¹, corresponding to the radial position $r = 2.36$ mm. **c** Same plot as (**b**) with a close-up between −20 and 20 s to focus on the stress relaxation and a log scale for the stress axis. **d** Collapse of the $(\sigma_{\theta z} - \sigma_{ss})/\sigma_{ss}$ as a function of $\Delta\gamma/\gamma_{ss}$. **e** Collapse of the same quantity for bubbles gaining contact during the same T1 event. **f** Stress relaxation duration $\Delta t_{T1}$, as measured in (**c**), as a function of the liquid fraction $\phi_\ell$. Error bars are obtained from statistics over all events.

contact maintains the same peak-to-peak stress drop amplitude of 40% (Fig. 4e).

In both cases, these variations highlight the intimate behaviour of the jammed bubbles which need first to deform before they can relax stress. The smooth continuous aspect of this dimensionless local stress-strain function can be attributed to the disordered structure of the bubble arrangement within our foams, which indeed appears very different from the sharp discontinuous functions obtained in simulations for ordered and crystalline bubble assemblies[40,41]. As such, this fine characterisation of the mechanical behaviour of the elementary structural elements of our soft jammed materials, (here the bubbles of a liquid foam) constitutes an important result, that will be particularly useful as a raw input in numerical simulations, and more generally for validating models of the yielding of liquid foams and other amorphous materials[42,43].

## Non-local quadrupolar stress redistribution of a T1 event
We can now extend our analysis further to investigate the stress variations that occur during such elementary topological rearrangements in their spatial surroundings (Fig. 5a).

First of all, a given T1 event is characterized by the directions of the lost and new contacts, which are represented by the unit vectors **a** and **b′**, respectively. These vectors are oriented along the lines connecting the centres of the corresponding bubbles (Fig. 5b). The orientation distributions of those two vectors, shown in Supplementary Information, display peaks in the $\theta z$ plane at directions close to ± 45° to the azimuthal and axial directions. These orientations correspond to the principal directions of the strain rate tensor in simple shear geometry, indicating that T1 events relax the elastic energy accumulated due to the imposed deformation.

For each T1 event, we define a local frame of reference {$O$, **a**, **b**, **c**} with $O$ marking the centroid of the four bubble centers involved in the T1, and (**a**, **b**, **c**) as the basis vectors defined as follows: **a** is the unit vector in the direction of the lost contact, as defined previously.

$\mathbf{b} = [\mathbf{b}' - (\mathbf{a} \cdot \mathbf{b}')\mathbf{a}]/[1 - (\mathbf{a} \cdot \mathbf{b}')^2]^{1/2}$ is the unit vector in the orthogonal direction to the lost contact in the plane (**a**, **b′**) (**b** is introduced because the directions of the lost contact **a** and of the new contact **b′** are never perfectly orthogonal). **c** = **a** × **b** is the unit vector in the direction orthogonal to the plane of the lost and new contacts. In this local frame, we can obtain a spatial map of the temporal stress variations occurring around a T1.

In practice, we compute the differences of the various components of the stress tensor $\Delta\sigma_{ij} = \sigma_{ij}(t+1) - \sigma_{ij}(t)$ (with $i$ and $j$ equal to $a$, $b$ or $c$), for two successive images during which a T1 event occurs, for each bubble whose centroid is located at a distance up to 8 bubble radius away from the T1 centre. We can then perform an average of those stress variations within 3D cubical zones (with a resolution of 25 × 25 × 25 voxels), measured over all the T1 detected at a given radial position (corresponding to a given local shear rate).

We consider our reference experiment corresponding to a dry foam (series 2 in Table 1), and include all the T1 events detected across all radii and times. The analysis is restricted to T1 events where all four bubble layers surrounding the T1 are fully captured, discarding events centered within the three bubble layers closest to the lower and upper plates. This filtering reduces the dataset to 13,300 events out of the original 33,000. The average steady-state shear stress, $\sigma_{ss} \simeq 35$ Pa, is used as the typical stress scale. Figure 5c–f display cross-sections of the 3D spatial maps of the stress redistribution around such elementary topological rearrangements. First of all, one can observe that a T1 event has a strong, non-local impact in its surroundings, with a non-trivial spatial perturbation of the stress field and specific angular dependencies in various directions. Indeed, the variation of the shear stress component $\Delta\sigma_{ab}$ displays a very clear quadrupolar pattern with increases and decreases alternating every eighth of a turn in the (a, b) plane. Given the orientation of (**a,b**), the poles of the quadrupolar shape are oriented at approximately ± 45° in the (θ,z) plane. On the other hand, the stress variation component $\Delta\sigma_{aa}$ tends to decrease, and the component $\Delta\sigma_{bb}$ to increase, along the lost contact direction $a$

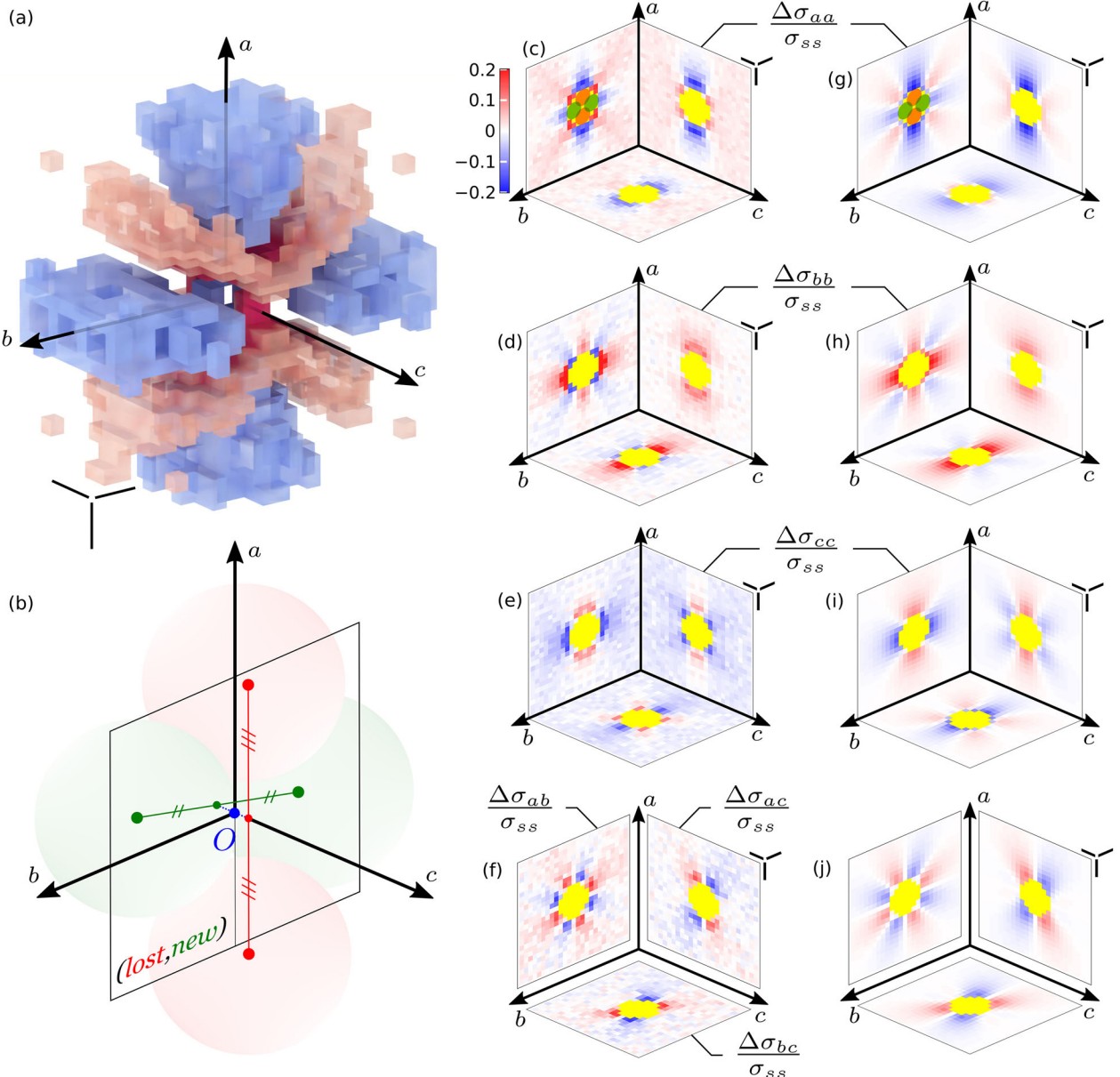

**Fig. 5 | Non-local quadrupolar stress redistribution of a T1 event. a** 3D representation of the average stress variation in the T1 frame of reference, using the example of the $\Delta\sigma_{aa}$ component, displaying a quadrupolar shape. The positive and negative deviations are displayed in red and blue, respectively, with transparency. **b** Definition of the T1 frame of reference {$O$, **a**, **b**, **c**} defined from the centers of the four bubbles involved in the T1 event. Bubbles losing and gaining contact are shown in red and green, respectively. **c**–**f** Average stress variations after the T1 event in the mid-planes ($a$, $b$), ($a$, $c$) and ($b$, $c$). Only non-zero cross-sections are shown.

**g**–**j** Average stress variations obtained using the model with the parameter $\varepsilon^s$ set to 0.068. In (**c**, **g**), the average positions and shapes of the four bubbles involved in the T1 events are depicted with the colors defined in (**b**). For comparison between experiments and model, we superimposed a spherical mask, in yellow, that marks the region encompassing the four bubbles involved in the T1 events. In all panels except (**b**), the three scale bars represent the mean bubble diameter in the three directions.

and new contact direction $b$, consistently with the main displacement of the corresponding bubbles across the T1. While the quadrupolar pattern was already reported in 2D[44,45], the full 3D picture appears much richer than the simple 2D case. For instance, the $cc$ component tends to increase in the $a$ direction, but to decrease in the $b$ direction. This means that the bubbles losing contact, respectively gaining contact, tend to relax by becoming less prolate, respectively less oblate. We furthermore observe clear angular patterns emerging in the two planes $ac$ and $bc$.

To go further, we can compare our measurements of the stress variations in the spatial surrounding of a T1 event to the ones obtained for an elastic medium due to an imposed quadrupolar deformation,

$\varepsilon^s = \varepsilon^s(\mathbf{a} \otimes \mathbf{a} - \mathbf{b} \otimes \mathbf{b})$. Indeed, such a deformation corresponds to the fact that the two bubbles losing contact get further apart while two bubbles gaining contact get closer. Thus, we assimilate the effect of a T1 as an imposed displacement $\mathbf{u}^s = \varepsilon^s \cdot \mathbf{x}$ at the surface of a sphere of radius $a$ centred on the T1. Considering that the foam reacts to this applied strain as an elastic Hookean material of Young's modulus $E$ and Poisson ratio $\nu = 1/2$ (the foam being quasi-incompressible), we can compare our measurements to Eshelby's prediction who solved this mechanical problem in a seminal work[32]. Note that foams are composed of discrete constituents—bubbles—whereas Eshelby's prediction applies to continuous media and is expected to be valid only on length scales larger than the typical bubble size. However, in this case, the

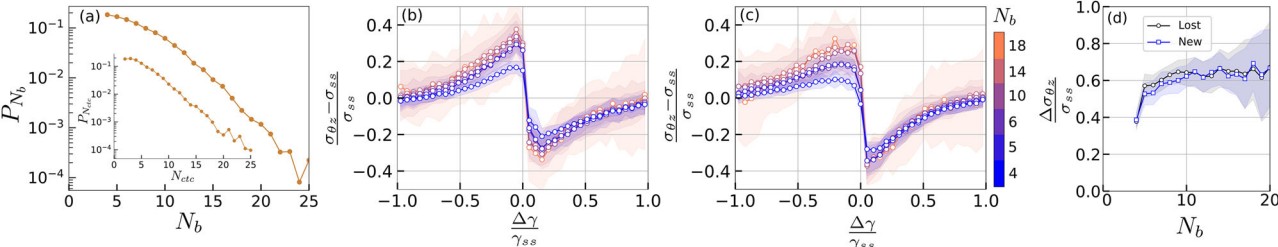

**Fig. 6 | Mechanical characterization of clusters of plasticity. a** Size distribution of the detected plastic clusters, involving $N_b$ connected bubbles, while shearing a dry foam, $\phi_l$ = 8.4%. The inset shows the distribution of the number of contact changes $N_{ctc}$ inside such clusters. The following panels provide stress-strain measurements -- averaged over various shear rates and liquid fractions -- showing

$(\sigma_{\theta z} - \sigma_{ss})/\sigma_{ss}$ as a function of $\Delta\gamma/\gamma_{ss}$, similar to (**d, e**) in Fig. 4, for bubbles either losing (**b**) or gaining (**c**) contacts, within a plastic cluster of size $N_b$.
**d** Corresponding peak-to-peak stress drops $\Delta\sigma_{\theta z}$, normalized by the steady-state shear stress $\sigma_{ss}$, as a function of the plastic cluster size $N_b$.

disordered structure of the foam and extensive averaging (thanks to our very large data-set) allow meaningful comparisons even at the scale of individual bubbles. We provide those predicted stress components in Supplementary Information. They share the common form $\Delta\sigma_{ij} = E\varepsilon^s f_{ij}(\mathbf{x}/a; \nu)$, where the dimensionless functions $f_{ij}$ quantify their spatial dependencies; notably, at large distances $x \gg a$, they decay as $1/x^3$. We use $\varepsilon^s$ as a single fitting parameter, for those various components $\Delta\sigma_{ij}$ to challenge such prediction. The agreement with our data is remarkable, including all the involved angular dependencies (see Fig. 5g–j for cross-sections of the 3D maps, and Supplementary Information for quantitative comparisons at specific positions). Furthermore, we could check that Eshelby's prediction is also verified for our various experimental parameters, e.g., at different radial position $r$ of the T1 and thus for different local shear rates $\dot\gamma$, and the various foams' liquid fractions $\phi_\ell$. While $\varepsilon^s$ appears independent of the local shear rates $\dot\gamma$, it decreases systematically with the liquid fraction (see Supplementary Information). This value should be correlated to the foam yield strain $\gamma_Y$ since we are computing the variations of the stress field over two successive images, with a time scale of 3 seconds, while a T1 rearrangement can last up to more than 10 seconds. For instance, for our reference experiment, we measure $\varepsilon^s = 0.068 \pm 0.002$. Considering the cumulative deformation over the duration of a T1 for such dry foam ( ~12 s), gives indeed a value close to $\gamma_Y \sim 0.26$.

Demonstrating that T1 rearrangements behave analogously to Eshelby inclusions is a significant result, as it helps explain how a single rearrangement can trigger others through long-range elastic interactions. This mechanism can potentially lead to avalanches and, more broadly, contributes to our understanding of the yielding behavior of soft amorphous materials. Overall, this constitutes the first experimental proof of the validity of the elastoplastic approach[42] for an amorphous material in 3D and directly on the local stress.

## Discussion

To conclude, we have demonstrated our ability to directly observe and monitor the 3D local displacements, stresses, and contact topology of the elementary microstructural components of a model soft jammed material under quasi-static shear flow. We could achieve such a challenging feat, thanks to the development of an innovative setup that leverages state-of-the-art 3D imaging, using time-resolved fast X-ray micro-tomography, combined with shear rheometry of well-chosen ultra-stable liquid foams. Through an accurate image analysis, our tomo-rheoscopy technique enables multi-scale stress measurements allowing us to establish links between the building blocks of these soft jammed materials (the bubbles) and the macroscopic intrinsic properties of the foams characterising their yielding behaviour. Indeed, we unveil the *universal* mechanical signature of the elementary topological rearrangement (involving the neighbour swapping of four bubbles), with a local stress-strain function independent of the foam's liquid fraction and the imposed local shear rates. Furthermore, we

demonstrate that such T1 plastic events redistribute stress non-locally and anisotropically, imposing quadrupolar deformations, within the foam behaving as an elastic medium.

It is important to note, however, that our analysis has so far focused on the simplest type of rearrangements. These so-called T1 events involve the loss of contact between two bubbles and the simultaneous formation of a new contact between two neighboring bubbles. In contrast, in our experiments, topological rearrangements can involve a significantly larger number of spatially connected bubbles, with multiple simultaneous contact formations and losses. Indeed, we have detected structural changes between consecutive images that indicate rearrangements involving many more linked bubbles than the four typically involved in a single T1 event. Figure 6a shows the size distribution of these plastic clusters for our reference shear experiment of a dry foam with a liquid fraction $\phi_l$ = 8.4%. We observe a broad distribution of plastic rearrangement sizes, with an exponential tail highlighted by the semi-log scale, involving up to more than 20 bubbles. The inset of this figure also shows the distribution of the number of contact changes (both losses and gains) $N_{ctc}$, which similarly exhibits a broad exponential tail. It should be noted that although T1 rearrangements account for only 20% of the various plastic events identified in such experiment, they represent the most frequently occurring type of rearrangement.

We have therefore extended our mechanical analysis to include these clusters of connected bubbles involved in the plastic activity of the sheared foams. Strikingly, panels (b, c) of Fig. 6 show that the stress-strain characteristics of these different plastic events represented by plots of $(\sigma_{\theta z} - \sigma_{ss})/\sigma_{ss}$ as a function of $\Delta\gamma/\gamma_{ss}$ –measured across various shear rates and liquid fractions– collapse on the same master curves as those observed for the T1 events. Notably, the corresponding peak-to-peak stress drops (shown in panel d) saturate at a value slightly above 60% for both contact losses and gains, regardless of the number of bubbles involved, although the measured stress variations appear slightly lower in the case of T1 events ($N_b$ = 4). Therefore, it appears reasonable to consider these plastic clusters as meta-T1 events, which should mediate stress redistribution in a manner similar to individual T1 rearrangements.

These observations of plastic rearrangements, which reveal strong spatial correlations in the bubble dynamics of sheared foams, warrant further analysis and experimentation. In particular, plastic clusters involving more than four bubbles may simply result from cascades of T1 events occurring within our current temporal resolution limit of 3s. Therefore, conducting new experiments with higher acquisition rates and over a broader range of applied shear rates would be highly valuable. This would allow for a more detailed investigation of the emergence of spatio-temporal correlations in bubble dynamics and facilitate the detection of avalanche-like sequences of plastic events. As such, the identification of large plastic clusters constitutes a critical initial step toward this objective.

## Methods

### Foam generation

The foaming solution was prepared by following the protocol of Golemanov et al.[46]. First, 6.6% sodium lauryl ether sulfate (SLES) and 3.4% cocamidopropyl betaine (CAPB) in mass were mixed in ultra-pure water; then 0.4% of myristic acid (MAc) was dissolved into the solution by stirring and heating at 60 ℃ for one hour; this solution was finally diluted 20 times with a glycerol/water mixture with a mass ratio of 50/50 to obtain the foaming solution. The gas used is a mixture of air and perfluorohexane, obtained by bubbling air through liquid per-fluorohexane. Since the presence of perfluorohexane is expected to alter the surface tension $\Gamma$[47], we measured the surface tension of the foaming solution in the presence of the gas mixture and found $\Gamma = 21.1 \pm 0.1$ mN.m$^{-1}$. Foams were then generated using a co-flow microfluidic setup by simultaneously introducing the foaming solution and the gas mixture. Precise control of solution flow rate and gas pressure was achieved using a Harvard Apparatus PHD Ultra syringe pump and a Fluigent MFCS-FLEX pressure controller, respectively. Subsequently, selected foams were subjected to centrifugation using an Eppendorf 5702 centrifuge at different rotation rates, to decrease their liquid fraction. In this paper, we focused on eight different foam samples, for which the liquid fraction $\phi_\ell$ and bubble size distributions (characterized through the Sauter mean radius $R_{32}$) were directly measured in situ from tomographic images (see the corresponding subsection *Image processing*) and reported in Table 1.

### Co-rotation rheometer for tomography

A dual-motor rheometer (Anton Paar MCR 702 TwinDrive) was adapted to allow for a co-rotation of the upper and lower plate geometries (to enable tomographic data acquisition) with a superimposed relative rotation rate (to produce shear), enabling precise application of controlled macroscopic deformation, torque measurement, and simultaneous 3D tomographic image acquisition. This innovative prototype, coined the *tomo-rheoscope*, was developed in a collaborative effort between PSI and the Laboratory of Food Process Engineering at ETH Zürich. The 2.75 mm radius plates were 3D printed and covered with sandpaper (P80). The foam was placed inside the rheometer, using a syringe. The gap $h$ was reduced to 1.5 mm, and the excess foam was removed from the outer free edge. The sample size corresponded to the available tomography field of view, allowing to image the whole sample during shearing.

### Tomography imaging

Imaging was performed at the TOMCAT beamline X02DA of the Swiss Light Source, Paul Scherrer Institute, Switzerland. X-rays generated by a 2.9T superbending magnet were monochromatized using a double-multilayer monochromator. High-energy X-rays (16 keV, 0.077 nm wavelength) were selected as a compromise between achieving a sufficient signal-to-noise ratio for phase-contrast image acquisition and minimizing the radiation dose to avoid altering the physical chemistry of the liquid foam. The total flux density in the monochromatic beam was approximately $10^{11}$ ph.s$^{-1}$.mm$^{-2}$. The rheometer holding the sample was placed 25 m from the source. Transmitted X-rays were detected 250 mm downstream from the sample using an X-ray microscopy system. The indirect detection setup included a 150 $\mu$m-thick LuAG:Ce scintillator screen to convert X-rays into visible light, a high numerical aperture 4 × magnification lens system, and a custom-designed fast CMOS detector with streaming data readout[48]. The detector sensor readout was cropped to 800 × 2016 pixels, covering a field of view of 2.2 × 5.5 mm$^2$ (v × h).

Each 3D image (or tomogram) was reconstructed from 1000 angular radiographic projections acquired over a 180 degrees rotation of the sample. The total exposure time per tomogram was 0.5 s, with the sample continuously rotated at 1 Hz. The raw projections were first corrected for background inhomogeneities and then phase-retrieved

using the single-distance homogeneous object approach[49,50]. These filtered projections served as input for tomographic reconstruction using a Fourier-based method[51,52], which directly interpolates between polar and Cartesian coordinates in Fourier space. A Parzen filter was integrated into the algorithm. Following each radiographic acquisition, a latency time of 2.5 s was introduced (with the shutter closed), yielding a tomogram acquisition rate of one every 3 s. This latency was determined based on the average expected bubble velocity at the sample periphery, optimizing the trade-off between minimizing X-ray exposure and capturing bubble dynamics effectively. Each dataset typically comprised 160 tomograms, each covering a volume of 2.2 × 5.5 × 5.5 mm$^3$, with a voxel size of 2.75 $\mu$m.

### Shear test protocol

To obtain comparable deformation history, each of the eight foam samples considered in this study were first sheared with a relative rotation speed of 50 mHz for 20 s in one direction followed by an equal duration in the opposite direction. After 80 s at 0 Hz, the foam was sheared at 1 mHz and simultaneously imaged for 480 s. Given the plate-plate geometry, the shear rate is a linearly increasing function of the radial position within the rheometer. The maximum value is reached at the periphery, with a value of 0.011 s$^{-1}$.

### Image processing

The 3D images were phase-segmented with an Otsu threshold. The binary values (0 and 1) correspond to the gas and liquid phases respectively. The images were then binned by joining eight neighbouring voxels and each individual bubble volume was segmented with the ITK watershed[53]. This step is crucial to correctly reconstruct the films between bubbles, which are too thin to be directly detected on the binary images. Physical properties at the scale of the foam were extracted and reported in the Table 1. For each bubble $i$, we obtain the volume $V_i$ and the equivalent radius $R_i = (3V_i/4\pi)^{1/3}$. From this, we obtain the average bubble radius for a given series as $R_{32} = \langle R_i^3 \rangle / \langle R_i^2 \rangle$, where the average is taken over all bubbles. Notice that our foams are almost monodisperse with standard deviation of the bubble radius distribution at least one order of magnitude lower than the average. We also checked that the distribution of bubble sizes remained spatially uniform throughout each experiment's duration. The liquid fraction $\phi_l = n_{liquid}/(n_{liquid} + n_{gas})$ is defined as the ratio of the number of voxels in the liquid phase to the total number of voxels in a phase segmented image, with $n_{liquid}$ and $n_{gas}$ the number of voxels in the liquid and gas phases respectively. The bubble volumes were extracted from the bubble segmented images. Typical cross-sectional zooms are shown in Fig. 7.

### Local velocity field measurements

The displacement of bubbles between two successive images is obtained by Discrete Digital Volume Correlation[53]. Bubble displacements are then averaged to infer velocity fields. An example of velocity measurements, $v_\theta$, for a given series is shown in Fig. 8a, with the convention that the velocity is zero at the contact with the lower plate. Plotting $v_\theta$ as a function of $zr$ reveals a proportionality relationship, as expected from the imposed boundary conditions, $v_\theta = \frac{\Omega}{h} zr$, with $\Omega = 2\pi \times 10^{-3}$ rad.s$^{-1}$ is the relative angular speed and $h = 1.5$ mm the gap between the rheometer plates.

### Mechanical characterization of the liquid foams

From the global stress-strain curves computed from our local image analysis, one can extract relevant rheological properties of our foam samples: the shear modulus $G$, the yield stress $\sigma_Y$, as well as the yield strain $\gamma_Y$. The yield stress is determined through a linear extrapolation of the function $\sigma_{ss}(\dot{\gamma})$ as $\dot{\gamma} \to 0$ (Fig. 8b). The yield strain $\gamma_Y$ is then calculated using the formula $\gamma_Y = \sigma_Y/G$. As shown in Fig. 9, their evolution with the liquid fraction $\phi_\ell$ is in quantitative agreement with data from the literature obtained from classical rheological measurements[37,38].

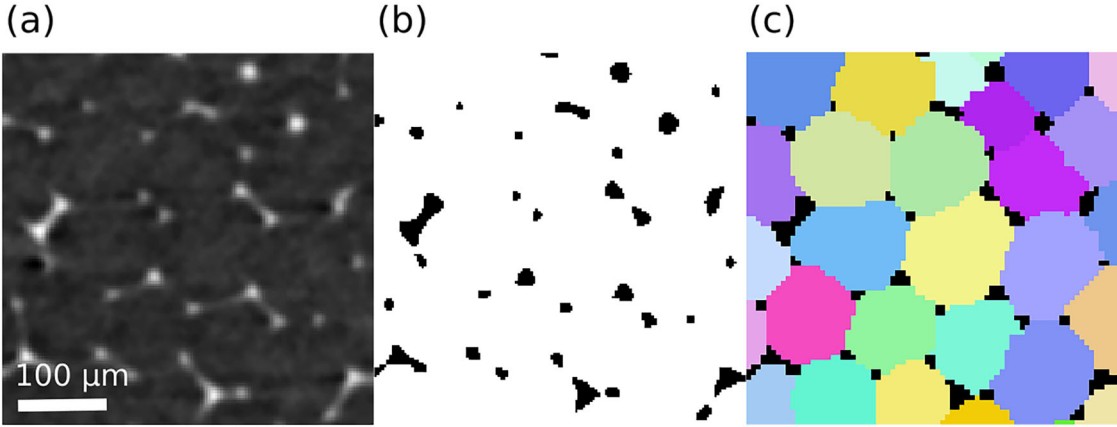

**Fig. 7 | Typical cross-sectional zooms from the same tomogram at different stages of the analysis (series 2 in Table 1). a** Original image, (**b**) binary image, and (**c**) segmented image.

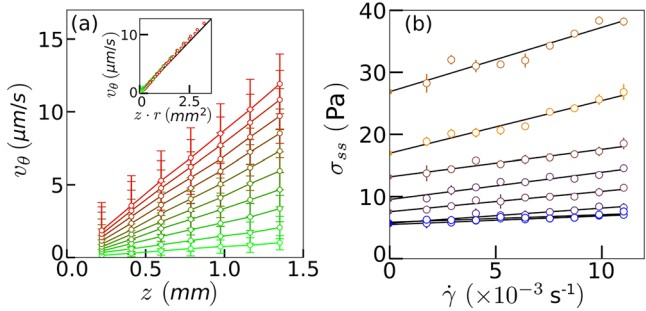

**Fig. 8 | Velocity and shear stress in steady conditions. a** Average velocity component $v_\theta$ as a function of height $z$ for various radii $r$ (series 2 in Table 1). Insert: $v_\theta$ as a function of $zr$. **b** Average steady-state shear stress $\sigma_{ss}$ inferred from local measurements as a function of the shear rate $\dot\gamma$ for the 8 series described in Table 1. The yield stress $\sigma_Y$ is obtained by linear extrapolation as $\dot\gamma \to 0$.

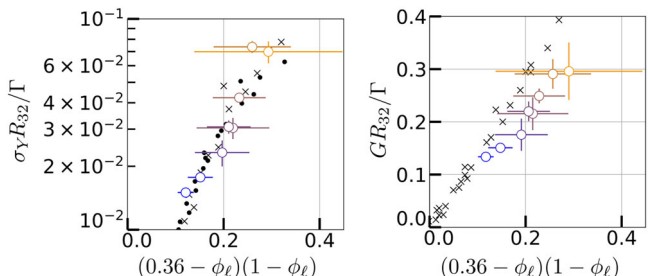

**Fig. 9 | Influence of the liquid fraction on the yield stress and shear elastic modulus of the foams.** Dimensionless yield stress $\sigma_Y$ and shear elastic modulus $G$ as functions of the liquid fraction, as proposed by Saint-Jalmes and Durian[38].

## Detection of plastic events

The bubbles within our liquid foams have a number of neighbors that decreases with the liquid fraction $\phi_l$, with the coordination number dropping from approximately 14 in dry foams ($\phi_l \simeq 6\%$), to around 11 in wet foams ($\phi_l \simeq 22\%$). During shear experiments, pairs of bubbles may either gain or lose contacts. The contact network at each time step is extracted using the Software for the Practical Analysis of Materials (SPAM)[53]. Specifically, we detect regions of overlap between bubbles after applying a one-pixel dilation. In contrast to the highly localized contacts investigated in assemblies of granular particles[54], the shared films between neighboring

bubbles in our foams are relatively extended, occupying a significant fraction of the bubble size. As a result, our analysis procedure is robust and less susceptible to errors from under- or over-detection of contacts.

We then define plastic events as topological changes in the contact network between neighboring bubbles occurring between successive images. An elementary plastic rearrangement, or T1 event, is identified when a new contact is formed (or lost) between the common neighbors of two bubbles that have lost (or gained) contact. If multiple contacts are gained or lost among adjacent bubbles, the detection procedure is applied iteratively until no further contact changes are observed within the set of involved bubbles. Each isolated plastic cluster is thus composed of connected bubbles that either gain or lose contacts. While a single T1 event involves four bubbles and two contact changes (one gained and one lost), larger plastic clusters encompass more bubbles and a greater number of contact changes. In our analysis, we classify plastic clusters according to the number of connected bubbles involved in topological changes, denoted $N_b$.

## Data availability

The raw data sets generated in this study are extremely large and therefore cannot be deposited in a public repository. However, a subset comprising two typical successive raw images from an experiment (series 2) is available at https://doi.org/10.5281/zenodo.15564270. In addition, down-scaled and cropped tomograms from the same experiment, along with documentation and a processing tutorial (Python package[34]), are available at foamquant.readthedocs.io. The data generated in this study are also provided in the Supplementary Information/Source Data file. Additional data can be obtained from the corresponding authors upon request. Source data are provided with this paper.

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

## Acknowledgements

We wish to thank the Swedish Research Council for funding this project (grant No. 2019-03742, F.S.). We acknowledge the Paul Scherrer Institut, Villigen, Switzerland for provision of synchrotron radiation beamtime at the TOMCAT beamline X02DA of the SLS. The Tomo-Rheoscope used in this study was funded by the Swiss National Science Foundation (Grant No. 205311, https://data.snf.ch/grants/grant/205311, C.M.S. and S.G.). S.S. thanks support from the CNRS and ENS de Lyon, through the IRP (D-FFRACT). We thank Landshövding Per Westlings Minnesfond, Sigfrid and Walborg Nordkvists, and Knut and Alice Wallenbergs Foundation for additional traveling found (grant No. RLh2023-0011, F.S.). We thank Thibaut Divoux, Sebastien Manneville, Emanuela Del Gado, Kirsten Martens, Vincent Démery, Alexis Poncet, Elisabeth Lemaire, Olga Volkova and Arthur Albarel for fruitful discussions.

## Author contributions

C.R., B.D., S.S. and R.M. designed the study. F.S., C.R., C.C., S.S., B.D. conceived the experiments with advice from S.G., C.M.S. and R.M.. C.C. and F.S. built up the microfluidic setup and produced the foams. S.G. and C.M.S. set-up the tomo-rheoscope. All the authors participated to the experimental campaign. F.S. performed the analysis under the supervision of B.D., S.S., C.R. and R.M.. S.S., C.R., B.D. and F.S. wrote the paper, and R.M., C.M.S., S.G. and C.C. provided editorial comments.

## Funding

## Competing interests

The authors declare no competing interests.
