## [Transparent Peer Review file · Nature Communications]

Multiscale stress dynamics in sheared liquid foams revealed by tomo-rheoscopy

Corresponding Author: Dr Florian Schott

Version 0:

Reviewer comments:

Reviewer #1

(Remarks to the Author)

Referee comments on Schott et al.

This is a tour-de-force experimental study of foam physics. The key component is the idea shown in Fig 2a. This allows to connect the physics on the scale of the details of bubble walls to the macroscopic. In spite of this, I have two types of comments that the authors should reply or react to before I can recommend publication.

The first concerns the scales of the experiment, I did not notice any comment on the bubble wall thickness vis a vis the imaging accuracy. Moreover I am confused. The timescale of the experiment is discretized by the imaging to steps of 3 seconds. However, several comments in the text seem not to take this into account (eg relaxation time being between 7 to 12 seconds).

The second category concerns the interpretation of the results and the claims. Let me start with a question: what type of YSF do these foams resemble? Herschel-Bulkley?

Second, the presence of such a viscous timescale with respect to T1 events has been claimed to be connected with the lack of clear avalanche activity in 2D foams (see Physical Review Materials 5 (7), 075601 and Phys. Rev. Research 2, 023338 (2020)).

These issues have a bearing on whether mesoscale elastoplastic models of now actually match foams.

The same question can be asked also as: how large stress fluctuations are there in non-T1 bubbles? If the local stress increases by 20 % before a T1 how much is not indicative still of a T1? 13 %? Presumably a typical bubble experiences stress variations with the same timescales, but what is the typical variation and how big is 20 % compared to that?

Last, the main point of the paper reading it with a tongue-in-cheek approach is hidden in the discussion, where the authors seem to say that well, the plasticity of these foams has to do with some other (what?) kind of events that are in a 3:1 majority. OK, maybe so but then the plastic activity is related to the redistribution kernels of those majority events so the Eshelby result while nice is not that important.

In other words the current claim of the field as far as I can see is that T1s are in foams the quanta of plasticity ("Shear-Transformation Zones") and the authors seem to indicate to the contrary. Apart from these considerations the conclusions part is not very well written.

PS note that Figure 2 caption is not correct. There are 4 subfigures.

Reviewer #2

(Remarks to the Author)

However I have some questions regarding the methodology. I think the answers will benefit a larger community than yours.

1) It is well known that liquid foams are sensitive to radiation damage.

a. How did you optimise the beamline parameters to prevent that?

b. It would be nice if you add details about the beamline parameters (insertion device and its parameters, presence or not of filters...), acquisitions parameters (number of projections), reconstruction parameters (phase retrieval algorithm and parameters...)

2) Data acquisition

a. Have you observed any effect of the rotation on the macroscopic rheological behaviour of the analysed foams?

b. Have you observed any blurr due sample shearing the tests? If yes, how did you cope that point to perform an accurate image analysis

c. Why did you bin the data and did not acquired them either in binning mode or at a larger pixel sizes ?

3) The image processing is a key point of the analysis and the results rely on it. I think this part would gain on visibility if there was a figure illustrating the accuracy of the segmentation. P6, you point out that there is bubble shape changes that I do not observe

Reviewer #3

(Remarks to the Author)

The paper presents a robust and unique experimental campaign that offers insights into stress measurements in liquid foam across scales, from individual bubbles to the entire sample. This is achieved using a novel rheometer prototype designed to fully exploit the spatial and, most importantly, temporal resolution of the X-ray tomography facility at the Swiss Light Source. The image analyses enable stress quantification at the micro level, which is then averaged over subregions and the whole sample, revealing remarkable agreement between the bubble-scale measurements and independent sample-scale ones. The paper is well written, introducing a novel approach for studying rheological properties from the micro to the assembly scale.

I recommend publication provided that the following points are addressed:

1 - To the best of my knowledge, the term "jamming" is used ambiguously in literature. Please define what is meant by "jammed."

2 - In the introduction, modify the sentence "3D image (a tomogram composed of thousands of projections)" to clarify that, in this case, a 3D image is the output of a reconstruction process.

3 - The authors compare their results with the Eshelby prediction. As I am not an expert in this area, I suggest the authors clearly justify how the Eshelby analytical solution applies to systems with multiple bubbles quite close to each other and the potential implications.

4 - Given the voxel size of 2.48 μm and a bubble diameter of 50 μm (approximately 20 voxels per diameter), the resolution is adequate for bubble tracking but it may be an obstacle to accurately estimating bubble surfaces. To ensure reproducibility, I would advice the authors to provide additional details on the methodology used to mesh the bubble surface and explain how the meshes compare to the expected bubble shape. This information could be included in the methods or supplementary material, clarifying whether the meshing is based on volume scaling, a marching cubes algorithm, or another techniques.

7 - The manuscript refers to "stress measurement," but since the stress is derived from image analysis rather than being directly measured, I suggest clarifying once this point in the manuscript.

8 - In Figure 2b, the evolution is said to reach a plateau, yet there seems to be a slight increase in $\sigma_{\theta z}$. It would be helpful if the authors could address whether this increase is within measurement sensitivity, related to variations in bubble counts across different radial intervals.

9 - The supplementary information should justify why the relative variation is considered "reasonably weak."

10 - I would decrease the transparency of the error bars in Figure 2c.

11 - A major point of results and discussion involves T1 events and their detection (the loss and gain of bubble contacts), yet it appears to me that the method for detecting these changes is not well described. It would be very beneficial to include details on how contacts are detected, perhaps by though the approach proposed by Wiebicke et al. (2017), given that the analyses have been carried out using the spam software. Based on the contact detection methodology, I would suggest to discuss the implications of potentially over- or under-detecting contacts, especially given that, from what I understand, T1 events are observed in some cases over a short duration only three sequential tomograms.

12 - Adding a visual example of bubbles deforming during a T1 event would support the reader.

13 - It is stated that bubbles can deform during a T1 event, yet the vectors 'a' and 'b' are defined based on the line connecting the centres of the involved bubbles. Could the authors explain how bubble deformation affects the definition of these vectors?

Version 1:

Reviewer comments:

Reviewer #1

(Remarks to the Author)

I studied the response and modifications provided the authors and I am now happy to recommend publications as it is.

Reviewer #3

(Remarks to the Author)

The authors have satisfactorily addressed all major points raised in my review. I have no further questions and recommend the manuscript for publication.

Reviewer 1

This is a tour-de-force experimental study of foam physics. The key component is the idea shown in Fig 2a. This allows to connect the physics on the scale of the details of bubble walls to the macroscopic. In spite of this, I have two types of comments that the authors should reply or react to before I can recommend publication.

We sincerely thank the Referee for their appreciative remarks, acknowledging the challenges involved in our work that we were able to overcome. Please find in the following our detailed point-by-point responses to the comments and questions.

The first concerns the scales of the experiment,

1. I did not notice any comment on the bubble wall thickness vis a vis the imaging accuracy. Moreover I am confused. The timescale of the experiment is discretized by the imaging to steps of 3 seconds. However, several comments in the text seem not to take this into account (eg relaxation time being between 7 to 12 seconds).

We thank the Referee for the relevant comments.

In the first version of our manuscript, we probably did not insist enough on the spatio-temporal scales (though it was notably specified in the Methods section), which indeed represent a clear challenge in our study.

The thicknesses of the bubble films of our liquid foams are smaller than the size of the voxel, which is $2.75 \mu\text{m}$ in our tomographic images. Therefore, while we have the spatial resolution to image the Plateau borders (and the vertices) whose typical size is a few tens of micrometers, we cannot directly observe and detect the bubble films (which are a few tens of nanometers thick). However, we have already developed and validated a phase and bubble segmentation and reconstruction procedure, through a watershed analysis (which we were able to improve in the present work) within a previous campaign and dedicated studies^{1,2}. The segmentation accuracy is now illustrated with an additional figure in the Methods section of the revised manuscript (see Fig. 1, here). While not a formal proof, the remarkable agreement between

Figure 1: Typical cross-sectional zooms from the same tomogram at different stages of the analysis: (a) original image, (b) binary image, and (c) segmented image.

our stress measurements derived from the shapes of the reconstructed bubbles and the independent torque values provided by the rheometer highlights the reliability and accuracy of our image analysis. More details are provided in our answer to the second and third referees.

Regarding the temporal resolution, the referee is pointing out relevantly one of the main limitations of our experiments. Indeed, we chose to record one 3D image (composed of 1000 2D projections acquired in 0.5 seconds) every 3 seconds, as a compromise between memory limitations and the targeted duration of the experiments (approximately 10 minutes), while limiting radiation exposure to avoid altering the liquid foam’s physical chemistry. Nevertheless, this limited temporal resolution is enough to characterize the temporal evolution of the topological rearrangements of our foam structure. Indeed, the characteristic time of the T1 events was defined arbitrarily as the duration of the detected stress drops, measured as the time interval between the maximum and minimum values in the time evolution of the local stress. It is worth noting that an alternative definition—such as the duration during which the absolute value of the local stress exceeds the typical background stress fluctuations—would have resulted in a slightly longer characteristic time scale. We measured an average characteristic time between 7.7 ± 1.6 and 11.5 ± 0.6 seconds, with error bars representing the standard deviation ($\pm\sigma$). This time corresponds to the duration of more than 2 to 4 consecutive recorded 3D-images. The very large dataset —comprising over 100,000 T1 events detected— ensures good statistical accuracy, despite the limited temporal resolution.

In the revised version of the manuscript, we have added few sentences to insist on the spatial and temporal scales of our experiments, and a figure in the Methods Section.

The second category concerns the interpretation of the results and the claims.

2. Let me start with a question: what type of YSF do these foams resemble? Herschel-Bulkley?

The liquid foams we formulated are yield stress fluids, with a flow curve following Herschel-Bulkley model $\sigma_{ss} = \sigma_Y + K\dot{\gamma}^n$.

Figure 2: Flow curves of our various liquid foams of different liquid fractions. Global measurement (black line) obtained in a plate-plate geometry, with 1.5 mm gap and 25 mm radius), on a wide range of shear rates $10^{-3} \text{ s} < \dot{\gamma} < 1$ shows that our liquid foams are yield stress fluids, that follow Herschel-Bulkley model $\sigma_{ss} = \sigma_Y + K\dot{\gamma}^n$, with an exponent n close to 0.25. Steady-state measurements obtained from our tomo-rheoscopy analysis at low shear rates, along with the extrapolated yield stress values (σ_Y , shown as colored circles), are superimposed.

We verified such a behaviour by performing new independent rheological measurements (plate-plate geometry, with 1.5 mm gap and 25 mm radius), on a wide range of shear rates 10^{-3}

$s < \dot{\gamma} < 1$ s over foams produced at various liquid fractions with the same formulation and the same micro-fluidic setup. Thus, we were able to generate similar liquid foams (with comparable liquid fraction and bubble size).

We show in Fig. 1, that indeed our liquid foams have a shear thinning behaviour with an exponent n close to 0.25, in quantitative agreement with the literature³. Furthermore, we superimpose on this figure the steady-state measurements obtained by our tomo-rheoscopy analysis at low shear rates (and the extrapolated values of the yield stress σ_Y), showing a quantitative agreement between our different and independent rheological measurements.

Those new complementary rheological measurements and corresponding figure are now included in the revised version of the manuscript in the Supplementary Information.

3. Second, the presence of such a viscous timescale with respect to T1 events has been claimed to be connected with the lack of clear avalanche activity in 2D foams (see Physical Review Materials 5 (7), 075601 and Phys. Rev. Research 2, 023338 (2020)). These issues have a bearing on whether mesoscale elastoplastic models of now actually match foams. The same question can be asked also as: how large stress fluctuations are there in non-T1 bubbles? If the local stress increases by 20 % before a T1 how much is not indicative still of a T1? 13 %? Presumably a typical bubble experiences stress variations with the same timescales, but what is the typical variation and how big is 20 % compared to that?

We thank the Referee for pointing out those interesting articles. The machine-learning tools developed to predict T1 events in those experiments appear very relevant. In future studies, we will certainly investigate such tools for analysing our very large 3D data sets. However, one has to be aware of the strong differences with our experiments. Indeed, in the references mentioned by the Referee, 2D foams - obtained with commercial dish washing solution and consisting of large bubbles, 4 mm diameter - were forced to flow radially in a circular Hele-Shaw cell at a high flow rate of 1 mm/s. Obviously our 3D geometry leads to much more complex structural features than in a 2D dry foam, for which a neighbour swapping event is easily and very simply defined. Furthermore, in 2D, wall friction plays a dominant role in bubble dynamics, modifying bubble interactions, the corresponding time scales of the flow, and the possible occurrence of avalanches.

As discussed in point 1, the characteristic timescale of the T1 events we measured –consistent with values reported in the literature⁴– can be linked to the interfacial properties of the bubble films, including their elasticity and viscosity. Indeed, we followed the formulation proposed by Golemanov et al⁵ to obtain rigid interfaces leading to long-lasting events, longer than the acquisition time of a tomograph (0.5 seconds), thereby limiting the risk of image blurring. This characteristic time scale is nevertheless significantly shorter than the timescale $1/\dot{\gamma}$ of the imposed deformation, making our shear experiments effectively quasi-static.

On the other hand, we could not find any clear discussion of “a viscous time scale with respect to T1 events that would be connected to a lack of clear avalanche activity” in those articles mentioned by the Referee. We are currently investigating the emergence of spatio-temporal correlations in the bubble dynamics across different types of experiments (including imposed shear flow as in the present study, and shear cessation tests to probe very different time scales). For instance, we are studying the distribution of waiting-times between subsequent plastic events. Preliminary results (shown in the figure below) indicate the emergence of a power-law distribution of waiting times, with an exponential cut-off (characteristic time) that diverges as the shear rate decreases, suggesting the development of temporal correlations

Figure 3: Distribution of waiting time between subsequent plastic events, as a function of the local shear rate (from series 3).

in the bubble flow. Such on-going work goes beyond the scope of the current manuscript. Nevertheless, as a first step towards the possible detection of avalanches of plastic events, we have now extended our mechanical analysis to include in the revised version of our manuscript, the various plastic events detected, not only those involving only four bubbles corresponding to the so-called T1 events. More details are given in our response to the following question.

Figure 4: Temporal variations of the stress values, measured around a T1 event, at the scale of a single bubble, and averaged over a larger number $N_b = 3$ up to $N_b = 10000$.

Finally, regarding the typical stress fluctuations experienced by the bubbles. We again thank the referee for the relevant comment that allowed us to clarify this important aspect of our

work. For a single bubble, the stress exhibits strong fluctuations as expected considering the disordered structure of the liquid foam. Those fluctuations can even be larger than the stress variation due to a T1 event. However, thanks to our extensive data sets, we can average our stress measurements over a large number of bubbles (10^4 to 10^5 per 3D image) or plastic events (of the order of 10^4), significantly reducing the standard error and thus the uncertainty around the average stress values. This important aspect of our measurements is illustrated in figure 4, where the stress level measured at the scale of a single bubble is averaged over a larger number of bubbles $N_b = 3$ up to $N_b = 10000$. Notably, this figure also highlights the challenge—if not the impossibility—of detecting individual plastic events by monitoring stress at the scale of a single bubble. However, a reliable estimate of the stress with reduced uncertainty becomes apparent once the stress is averaged over approximately 100 bubbles

This important aspect of our stress measurement and corresponding figure are now included in the revised version of the manuscript in the Supplementary Information.

4. Last, the main point of the paper reading it with a tongue-in-cheek approach is hidden in the discussion, where the authors seem to say that well, the plasticity of these foams has to do with some other (what?) kind of events that are in a 3:1 majority. OK, maybe so but then the plastic activity is related to the redistribution kernels of those majority events so the Eshelby result while nice is not that important. In other words the current claim of the field as far as I can see is that T1s are in foams the quanta of plasticity (“Shear-Transformation Zones”) and the authors seem to indicate to the contrary. Apart from these considerations the conclusions part is not very well written.

We thank the Referee for these relevant observations, which constitute a direct follow-up and a reformulation of their previous comments (point 3). We acknowledge that the main message of the paper may not have been sufficiently clear. To address this, we have included new materials and revised the discussion/conclusion section to better clarify and highlight our key results and conclusions, ensuring that the main results are communicated more explicitly and without ambiguity. We would like indeed to clarify a misunderstanding, since we do not want to carry out an eventual misleading message that would state that the plasticity of [our sheared] foams [arise from] other type of events that than the typical T1 events, that cannot be described by current elasto-plastic models.

In the first version of the manuscript, we focused on the simplest type of rearrangement events. These so-called T1 events with a neighbor exchange between four bubbles, where two bubbles lose a contact while two others gain one, are clearly defined and observed in 2D dry foams. However, for 3D wetter foams, the topological rearrangements often involve a larger number of spatially connected bubbles, with numerous simultaneous contact formations and losses, making their characterization complex.

Nevertheless, we have now extended our analysis to include the various corresponding clusters of contiguous bubbles involved in topological rearrangements, detected between two successive time steps, and not just those involving only four bubbles. We classify these events according to the number of connected bubbles involved.

For instance, the panel (a) of figure 5 shows the size distribution of those plastic rearrangements for our reference experiment on a dry liquid foam. We observe a broad size distribution with an exponential tail, with plastic events involving up to 25 bubbles. Similarly, the number of contact changes follow an exponential distribution. One should notice that

Figure 5: **Mechanical characterization of clusters of plasticity** –

(a) Size distribution of the detected plastic clusters, involving N_b connected bubbles, while shearing a dry foam ($\phi_l = 8.4\%$). The inset shows the distribution of the number of contact changes N_{ctc} . The following panels provide stress-strain measurements – averaged over various shear rates and liquid fractions – showing $(\sigma_{\theta z} - \sigma_{ss})/\sigma_{ss}$ as a function of γ/γ_{ss} , for bubbles either loosing (b) or gaining (c) contacts, within a plastic cluster of size N_b . (d) Corresponding stress drops $\Delta\sigma_{\theta z}$, normalized by the steady-state shear stress σ_{ss} , as a function of the plastic cluster size N_b .

T1 rearrangements are the most frequent events. Furthermore, these plastic clusters involving more than four bubbles may simply result from cascades of T1 events occurring within our limited time resolution of 3 seconds. Strikingly, the stress-strain characteristics of these various plastic events follow the same master curve. Interestingly, the corresponding stress variations (increases and drops) with respect to the steady-state value saturate around 60% for contact losses and 50% for contact gains, independently of the number of bubbles involved, although the measured stress values appear slightly lower for the T1 events. Therefore, we can reasonably consider these plastic clusters as meta-T1 events, which should mediate stress redistribution in a manner similar to individual T1 rearrangements.

We have incorporated these new results and corresponding figures into the revised version of the manuscript, in the “Discussion” section.

5. PS note that Figure 2 caption is not correct. There are 4 subfigures.

We thank the Referee for their careful reading, we have now corrected the caption of Fig. 2.

References

- [1] K. Mader, R. Mokso, C. Raufaste, B. Dollet, S. Santucci, J. Lambert and M. Stampanoni, Colloids Surf. A, 2012, **415**, 230–238.
- [2] C. Raufaste, B. Dollet, K. Mader, S. Santucci and R. Mokso, EPL, 2015, **111**, 38004.
- [3] N. D. Denkov, S. Tcholakova, K. Golemanov, K. P. Ananthpadmanabhan and A. Lips, Soft Matter, 2009, **5**, 3389–3408.
- [4] A. L. Biance, S. Cohen-Addad and R. Höhler, Soft Matter, 2009, **5**, 4672–4679.
- [5] K. Golemanov, N. D. Denkov, S. Tcholakova, M. Vethamuthu and A. Lips, Langmuir, 2008, **24**, 9956–9961.

Reviewer 2

However I have some questions regarding the methodology. I think the answers will benefit a larger community than yours.

1. It is well known that liquid foams are sensitive to radiation damage.

- (a) How did you optimise the beamline parameters to prevent that?

Indeed the aim was to reduce as much as possible the radiation dose that could affect the liquid foam physical chemistry. High energy X-rays (16 keV, 0.077 nm wavelength) were selected for their weak absorption interaction with matter, while their high spatial coherence was exploited for phase contrast to obtain images of adequate signal to noise ratio with the minimum possible exposure time. We invested considerable effort in finding the optimal balance between X-ray energy and exposure time to ensure acceptable image quality. Through systematic testing, we identified 16 keV as the most suitable energy. While, in theory, higher X-ray energies could further reduce absorption (and thus the radiation dose), in our setup, absorption at 16 keV was already well below 10%. Increasing further the X-ray energy did not lead to a meaningful reduction in dose, but rather degraded image quality due to the weaker phase-contrast signal resulting from reduced X-ray–matter interaction.

We have added a description in the Methods section.

- (b) It would be nice if you add details about the beamline parameters (insertion device and its parameters, presence or not of filters...), acquisitions parameters (number of projections), reconstruction parameters (phase retrieval algorithm and parameters...)

We took into account the reviewer comment, and have now added details about the beamline parameters in the Methods section.

2. Data acquisition

- (a) Have you observed any effect of the rotation on the macroscopic rheological behaviour of the analysed foams?

The tomography acquisition was done at a $f = 1$ Hz rotation frequency with a sample having a 5.5 mm diameter. At a given radius r , the foam experiences centrifugal acceleration $4\pi^2 f^2 r$, but it remains below 1 m/s^2 within the observation window, hence negligible compared to gravity. We moreover double checked a possible centrifugal effect along the radial direction on the liquid fraction distribution, but it was not detectable (see Fig. 3b in the Supplementary Information).

We have added a comment in the Supplemental Information, while discussing Fig. 3b.

- (b) Have you observed any blur due sample shearing the tests? If yes, how did you cope that point to perform an accurate image analysis

We did not observe any noticeable blur in our images. The rotation speed was chosen to ensure that bubble motion driven by shear during the 0.5 s exposure time was minimal, while only slight movement occurred during the 2.5 s latency period. In addition, the relaxation time of T1 events (of the order of 10 s) is much longer than the exposure time, ensuring that no significant motion occurs during the 0.5 s required to acquire each tomogram.

We have added details in the Methods section, tomography imaging.

- (c) Why did you bin the data and did not acquired them either in binning mode or at a larger pixel sizes ?

We wanted the best spatial resolution corresponding to the given field of view. During the processing, we worked with full resolution volumes up to and including the bubble segmentation step. The liquid fraction is therefore based on full resolution images. However, we faced a computing limitation when analysing our time-resolved experiments and decided to bin $2 \times 2 \times 2$ to lower the analysis pipeline from a day to an hour per tomogram. This choice was verified on several occasions by comparing the results obtained on full resolution and binned volumes when we were devising our image analysis workflow.

3. The image processing is a key point of the analysis and the results rely on it. I think this part would gain on visibility if there was a figure illustrating the accuracy of the segmentation.

We thank the reviewer for this comment. The voxelization of the bubbles and the accuracy of the segmentation are now illustrated in an additional figure in the Methods section (see Fig. 1 here or Fig. 7 of the revised manuscript), showing a zoomed cross-sectional view of the foam using the same colour code as in Fig. 1c. To further demonstrate the quality of the segmentation dynamically, we have added a movie of a tracked T1 event in the Supplementary Information.

Figure 1: Typical cross-sectional zooms from the same tomogram at different stages of the analysis: (a) original image, (b) binary image, and (c) segmented image.

4. P6, you point out that there is bubble shape changes that I do not observe

Following the referee's suggestion, we have added a figure showing an example of a T1 event, where the deformation of the bubbles can be observed.

Reviewer 3

The paper presents a robust and unique experimental campaign that offers insights into stress measurements in liquid foam across scales, from individual bubbles to the entire sample. This is achieved using a novel rheometer prototype designed to fully exploit the spatial and, most importantly, temporal resolution of the X-ray tomography facility at the Swiss Light Source. The image analyses enable stress quantification at the micro level, which is then averaged over subregions and the whole sample, revealing remarkable agreement between the bubble-scale measurements and independent sample-scale ones. The paper is well written, introducing a novel approach for studying rheological properties from the micro to the assembly scale.

I recommend publication provided that the following points are addressed:

We sincerely thank the Referee for their appreciative remarks. Please find in the following our detailed point-by-point responses to the comments and questions.

1. To the best of my knowledge, the term “jamming” is used ambiguously in literature. Please define what is meant by “jammed.”

In the introductory part of the manuscript, we refer to foams as an archetypal example of soft jammed materials, with the bubbles as their elementary “jammed” constituents. Indeed, when the bubbles are densely packed and trapped by their neighbours, the foam behaves as a rigid elastic solid, capable of sustaining and transmitting forces through their structure. The liquid fraction of the liquid foams controls their mechanical behaviour. Above a critical liquid fraction ϕ^* , the foam transitions into a bubbly liquid, with the mean number of neighbouring bubble dropping below the isostatic threshold—the minimum number of contacts required to maintain mechanical stability. The bubble network can no longer support elastic stresses.

We have added a sentence in the manuscript to clarify what we mean by “jammed”.

2. In the introduction, modify the sentence “3D image (a tomogram composed of thousands of projections)” to clarify that, in this case, a 3D image is the output of a reconstruction process.

We have followed the reviewer’s recommendation and revised the sentence for clarity. It has been updated to: “3D image (a tomogram reconstructed from thousands of 2D projections)”.

3. The authors compare their results with the Eshelby prediction. As I am not an expert in this area, I suggest the authors clearly justify how the Eshelby analytical solution applies to systems with multiple bubbles quite close to each other and the potential implications.

The Eshelby inclusion problem is a classic problem in continuum elasticity theory that examines how an inclusion – a region with different mechanical properties or undergoing a transformation strain – embedded in an infinite elastic medium influences the surrounding stress and strain fields. It plays a fundamental role in understanding stress redistribution in heterogeneous materials.

However, while this theoretical framework provides a powerful tool for analyzing how local rearrangements affect the macroscopic mechanical response of soft amorphous materials, such as foams, the Reviewer rightly pinpoints a significant challenge in comparing Eshelby’s predictions to our system. While the former hold for an elastic continuum, the T1 event only implies four bubbles, and the stress variations are computed for a rather small number of bubbles in the surrounding of each T1, in apparent contradiction with the requirement of a

large number of microstructural units for a continuum approach. Actually, the key is the averaging over many (several thousands) of T1 events, which provides enough smoothness to test Eshelby’s predictions. Moreover, our foams are disordered enough that we are not influenced by anisotropy or crystallinity in the surrounding of such events. This is what validates the Eshelby approach in our case. Notice that such an approach has also already been applied with success to granular system¹ despite the same issues with the finite size of the constituents.

Here, we show that T1 rearrangements behave analogously to Eshelby inclusions. This result is significant because it helps explaining how a single rearrangement can trigger others through long-range elastic interactions, potentially leading to avalanches, and, more generally, to understand the yielding behavior of soft amorphous materials.

In the revised version of the manuscript, We added two distinct paragraphs to clearly justify this approach and the implications of the corresponding results.

4. Given the voxel size of 2.48 μm and a bubble diameter of 50 μm (approximately 20 voxels per diameter), the resolution is adequate for bubble tracking but it may be an obstacle to accurately estimating bubble surfaces. To ensure reproducibility, I would advice the authors to provide additional details on the methodology used to mesh the bubble surface and explain how the meshes compare to the expected bubble shape. This information could be included in the methods or supplementary material, clarifying whether the meshing is based on volume scaling, a marching cubes algorithm, or another techniques.

We agree with the reviewer. This point indeed required clarification. Please note that there was a mistake concerning the pixel size. It is equal to 2.75 μm (text and figures updated). The mesh was done by a simple marching cube algorithm (scikit-image). Indeed the mesh presents some degree of noise compared to the expected bubble shape because of the voxelization and potential watershed segmentation artefacts. Although some methods are in principal possible to smoothen the surface mesh obtained by the marching cube algorithm (e.g. the Catmull–Clark subdivision), which may improve the measurement of stress, we did not implement them because of their large computation time. See the example of meshed bubble surface with the marching cube:

Figure 1: Marching cube meshing of the bubble voxelized surface.

Individual bubble stress σ_{ij} was measured from full resolution images (bin1), and compared to the same stresses measured from images with a binning 2, 3 and 4 (binning i means

a reduction of a factor i in each direction, hence only one voxel over i^3 is retained). This comparison includes therefore the effect of the watershed and marching cube processing steps. We observe a robust measure of stress that depends very little on the image resolution. With our measures (binning 2) we show that the bias compared to the full resolution is at the largest 6% of the stress measure, for the wettest foam, with the lower measured stress amplitudes. The noise introduced by the binning 2 ($\pm 1\sigma$) was estimated to be at largest equal to 8.0 ± 0.4 Pa, this in the dry case, since there is a larger thin-films area fraction that should be reconstructed by the watershed algorithm and therefore induces larger fluctuations.

Figure 2: Series 2: Individual bubble stress components σ_{ij} obtained from binned images (bin2, bin3, bin4) as a function of the full resolution stress measure. The slopes a for the fit $y = ax$ were respectively equal to 0.985, 0.981 and 0.960 for the comparison with bin2, bin3 and bin4.

Figure 3: Series 8: Individual bubble stress components σ_{ij} obtained from binned images (bin2, bin3, bin4) as a function of the full resolution stress measure. The slopes a for the fit $y = ax$ were respectively equal to 0.944, 0.912 and 0.890 for the comparison with bin2, bin3 and bin4.

- The manuscript refers to “stress measurement,” but since the stress is derived from image analysis rather than being directly measured, I suggest clarifying once this point in the manuscript.

We have followed the reviewer’s suggestion. In the revised version of the manuscript, we now insist on this aspect, writing notably in the introduction, “... thus validating our local stress measurements, derived from image analysis of the reconstructed bubble film shapes”.

- In Figure 2b, the evolution is said to reach a plateau, yet there seems to be a slight increase in $\sigma_{\theta z}$. It would be helpful if the authors could address whether this increase is within measurement sensitivity, related to variations in bubble counts across different radial intervals.

There is indeed a slight increase of the steady state stress σ_{ss} of the θz component with time. This can be directly attributed to the slight decrease of the liquid fraction over the 8 min course of the experiments as already mentioned in the first version of the manuscript. Indeed, this slight variation is consistent with the overall trend of the steady-state shear stress as a function of the liquid fraction ϕ_ℓ across all experiments, which were conducted on liquid foams spanning a wide range of liquid fractions, as shown in the figure below. Furthermore, we verified that the number of bubbles remained constant throughout the experiments.

Figure 4: $\sigma_{ss} R_{32} / \Gamma$ as a function of the liquid fraction at various shear rates (color code) and for all the experiments. The tendency in each experiment is coherent with the global trend throughout all the series.

- The supplementary information should justify why the relative variation is considered “reasonably weak.”

We chose to remove this unnecessary and subjective remark, and now present only our quantitative measurement of the relative variation in liquid fraction.

- I would decrease the transparency of the error bars in Figure 2c.

We thank the referee for this advice to increase the visibility of the figure. We have modified the transparency accordingly.

9. A major point of results and discussion involves T1 events and their detection (the loss and gain of bubble contacts), yet it appears to me that the method for detecting these changes is not well described. It would be very beneficial to include details on how contacts are detected, perhaps by though the approach proposed by Wiebicke et al. (2017), given that the analyses have been carried out using the spam software. Based on the contact detection methodology, I would suggest to discuss the implications of potentially over- or under-detecting contacts, especially given that, from what I understand, T1 events are observed in some cases over a short duration only three sequential tomograms.

We have followed the relevant suggestion of the reviewer. In the revised version of the manuscript, we have included in the “Methods” Section, a paragraph dedicated to the detection of topological rearrangements.

Indeed, during our shear experiments, pairs of bubbles may either gain or lose contacts. The contact network at each time step is extracted using the Software for the Practical Analysis of Materials (SPAM)². Specifically, we detect regions of overlap between bubbles after applying a one-pixel dilation. Even though such a morphological operation leads to an overestimation of the contact size, we can nevertheless be confident in the number of detected events. Indeed, in contrast to the localized contacts observed in granular particle assemblies³, the shared films between neighboring bubbles in foams are extended, occupying a significant fraction of the bubble size. (typically a diameter of 10 voxel-size $\simeq 30 \mu\text{m}$).

As a result, our analysis procedure is robust and less susceptible to errors from under- or over-detection of contacts.

10. Adding a visual example of bubbles deforming during a T1 event would support the reader.

We agree with the reviewer. Therefore, we have included new figures in the revised manuscript, both in the article and in the Supplementary Information, that visually illustrate the deformation of the four bubbles involved in a T1 event at different time steps. Furthermore, the corresponding video has been provided as Supplementary Information.

11. It is stated that bubbles can deform during a T1 event, yet the vectors ‘a’ and ‘b’ are defined based on the line connecting the centres of the involved bubbles. Could the authors explain how bubble deformation affects the definition of these vectors?

We thank the referee for this insightful comment. We chose to construct the orthonormal basis (\mathbf{a} , \mathbf{b} , \mathbf{c}) using the positions of bubble centers, as these are robust quantities whose definitions are independent of bubble deformation and evolve smoothly over time, even during a T1 event. This approach allows us to track how the basis is advected throughout the T1 event and to compute the resulting changes in the stress field. We preferred this method over using the normal vectors of the appearing or disappearing films, which are more sensitive to noise and cannot be consistently advected.

References

- [1] A. Le Bouil, A. Amon, S. McNamara and J. Crassous, Phys. Rev. Lett., 2014, **112**, 246001.
- [2] O. Stamati, E. Andò, E. Roubin, R. Cailletaud, M. Wiebicke, G. Pinzon, C. Couture, R. C. Hurley, R. Caulk, D. Caillerie, T. Matsushima, P. Bésuelle, F. Bertoni, T. Arnaud, A. Ortega Laborin, R. Rorato, Y. Sun, A. Tengattini, O. Okubadejo, J. B. Colliat, M. Saadatfar, F. E. Garcia, C. Papazoglou, I. Vego, S. Brisard, J. Dijkstra, and G. Birmipilis, J. Open Source Software, 2020.
- [3] M. Wiebicke, E. Andò, I. Herle and G. Viggiani, Measurement Science and Technology, 2017, **28**, 124007.